# Which Food Outlets Are Important for Nutrient-Dense-Porridge-Flour Access by the Base-of-the-Pyramid Consumers? Evidence from the Informal Kenyan Settlements

Kevin Kipkemei Koech [1],*, Christine G. Kiria Chege [2],*  and Hillary Bett [1]

[1]   Department of Agricultural Economics and Agribusiness Management, Egerton University, Njoro P.O. Box 536-20107, Kenya
[2]   International Centre for Tropical Agriculture (CIAT), Nairobi P.O. Box 823-00621, Kenya
*   Correspondence: koechkevin39@gmail.com (K.K.K.); c.chege@cgiar.org (C.G.K.C.)

**Abstract:** Many Kenyan base-of-the-pyramid (BoP) consumers—defined as the poorest two-thirds of the economic human pyramid—remain food insecure, despite the availability of nutrient-dense foods in the market. This study reveals how effective marketing strategies can strengthen food security among BoP consumers through increased access to nutritious foods such as nutrient-dense porridge flour. Nutrient-dense porridge flour refers to a multi-composite porridge flour composed of diverse nutritious ingredients that are necessary to achieve a healthy diet. The main objective of the study was to determine the most effective channel for making nutrient-dense porridge flour available to BoP consumers. Data were collected through a cross-sectional survey in Kawangware, Nairobi County, using a multistage sampling design and a sample size of 603, via structured questionnaires. A multivariate Probit (MV-Probit) model was used to analyze the most effective channel for providing nutrient-dense flour to BoP consumers. The study results show that the most common outlets used to access this flour were supermarkets (51.08%), followed by cereal stores (25.54%). According to previous studies, using appropriate marketing strategies leads to increases in the uptake and consumption of nutritious products in informal urban settlements among developing countries. Consequently, policies and interventions targeting BoP consumption of nutritious products toward reducing food and nutrition insecurity in informal urban settlements should be based on appropriate marketing strategies that consider the institutional factors and significant household characteristics of the BoP communities.

**Keywords:** base-of-the-pyramid consumers; multivariate probit; food insecurity; effective outlets; informal urban settlements

## 1. Introduction

Global food production is abundant/sufficient, yet approximately 0.8 billion people still go hungry [1]. One of the major global challenges is that of ensuring sufficient quantities and quality of food to cater for the nutritional needs of an ever-growing population, which is projected to increase to about 10 billion by the year 2050 [2].

Progress toward improving food security continues to be uneven across Sub-Saharan Africa regions [3]. Remarkable progress toward reducing hunger has been made by some regions, such as Northern Africa, Latin America, Eastern Asia, and Central Asia [4]. Progress in Sub-Saharan Africa (SSA) and Southern Asia, however, has been poor, further compacting/increasing to the point of creating major food-insecurity crises in these regions [1].

The individuals who are most vulnerable to food insecurity in Kenya are those living in the arid and semi-arid areas and the informal urban settlements, also referred to as slums [5]. About a quarter of the Kenyan population lives in these areas, where disease,

conflict, structural underdevelopment, and poverty are rampant [6]. Consumers living in informal urban settlements have been categorized as the base-of-the-pyramid consumers (BoP). These are the resource-poor consumers living on less than US $1.9 a day [7,8].

The prevalence of food insecurity in Kenya has been attributed to unsupportive political activities, environmental factors (e.g., rainfall patterns), and socioeconomic factors [1]. The number of individuals living under acute food insecurity declined to 2.6 million in 2018 from 3.4 million in 2017 [3]. This improvement has been attributed to increased rainfall, widespread nutrition response mechanisms by governmental and non-governmental agencies, emergency cash and food transfers, and early diagnosis and treatment of acute malnutrition [5].

Dietary diversification—in this context, meaning that households and individuals have access to a wide variety of food items in their daily meals—is one of multiple approaches that can be used to reduce malnutrition [9]. Improved nutrition could be made possible if frequently consumed foods are diversified so that they contain more than one food item or, rather, foods from more than one food group [10]. One of the foods that is mostly consumed by the BoP consumers in the informal urban settlements is soft porridge, and it is consumed as a complementary food for young children, by pregnant and lactating mothers, and by the other household members either in the morning, as a snack, or in the evening [4]. Most of the porridge consumed by BoP consumers in the informal settlement is not diversified. Most often, the porridge is cereal-based, and it contains only one food item, such as maize or millet, or a combination of the two, which eliminates hunger but lacks important micronutrients that are needed to support optimal physical growth and mental development. Addressing hunger alone is not sufficient [11]. There is an urgent need to address micronutrient deficiency. This would entail diversifying porridge flour to include other food items that will provide important micronutrients to the consumers [12]. However, it should be noted that diversifying the porridge flour may increase its cost, so that some of the BoP consumers may not be able to afford it [8].

Affordability, accessibility, and availability are important factors determining whether or not diversified nutrient-dense porridge flour would be purchased by the target consumers [2]. Affordability increases the consumers' purchasing power and thus increases the quantity of porridge consumed. Accessibility and availability both reduce the transaction costs linked to information and access, thus indirectly increasing nutrient-dense-porridge-flour consumption. Availability indicates the extent to which nutritious foods are stocked in the right market channels that are common among BoP consumers, while accessibility refers to the process of ensuring that these foods can be easily reached physically by the consumers based on the location of the outlets.

Increasing access to affordable and high-quality diets via value chains meant to improve nutrition is an effective way of dealing with malnutrition, which is highly prevalent among the BoPs [13]. Furthermore, food preparation time will also determine whether and how often these foods are consumed. Foods that can be cooked quickly are highly demanded since they save on resources (cooking fuel) and cooking time, thus allowing the poor consumers more time for other activities, such as work [14].

Most often, the BoP consumers have limited access to nutrient-dense foods because the foods are developed, introduced, priced, distributed, and marketed in ways that do not often consider the social and economic circumstances of these consumers [15]. This makes it difficult for these consumers to access the foods despite their presence in the market. To help address this challenge, the main objective of this study was to assess the consumption of nutrient-dense porridge flour by BoP consumers in the informal settlements of Kawangware in Nairobi. Kawangware dwellers are classified as BoP consumers since they live on less than US $1.9 a day and they are not able to afford more than one meal a day [16]. It is important to assess the purchasing power of these consumers by considering available market outlets, to ascertain the factors that influence their purchasing decisions. This will help to inform the right food-marketing strategies in the informal urban settlements.

Globally, food abundance exists, yet millions of people especially in the informal urban settlements of developing countries are still food insecure [17]. This can be largely attributed to factors such as accessibility and low purchasing power among potential consumers. Over 10% of the total food retailing in Kenya is accounted for by supermarkets, and more than 20% of food retailing in large cities in Africa is also accounted for by the same market channel [18]. Information regarding the contribution of other market distribution channels, such as kiosks and mom-and-pop shops, is not yet clear. There is a need to ascertain the role of market access and household characteristics in achieving food security, especially in informal urban settlements. The right market channels and product-promotion strategies need to be used when availing nutrient-dense food to the BoP's, and this will help in dealing with the current situation of food insecurity in the informal urban settlements.

Even though traditional distribution channels are regarded as being complex, they are the most effective channels to reach the BoP [19]. Traditional channels, in this case, include kiosks and door-to-door delivery methods. This report argues that traditional channels can build customer loyalty and create demand. On the other hand, it is also argued that the use of supermarkets among developing countries has been known to have a positive correlation with the income of households [20,21]. The information on the most effective outlet that can be used to reach out to the BoP is thus not clear; hence, it is imperative to carry out more research.

This study conducted an assessment to ascertain the best market channels that can be used to supply the products to the consumers. So far, little is known regarding the best marketing channels and product-promotion strategies that can be used in such informal urban settings. A total of 603 respondents were interviewed for this study. The study placed emphasis not only on the entire households but also on women of reproductive age (15–49 years) and children under five years of age (6–59 months). The latter are the most vulnerable in terms of attaining a healthy and nutritious diet [22].

## 2. Materials and Methods

### 2.1. Study Area

This study was carried out in Kawangware, Nairobi County. Kawangware is an informal urban settlement, or slum area, which lies approximately 15 km to the west of the Nairobi Central Business District as shown in Figure 1. It extends between longitude 36°44′37″ E and latitude 1°17′4″ S. Kawangware occupies an area of 1.2 square kilometers and is among the fastest-growing slums in Nairobi [23]. It has a total population of 133,286, of which 65% of the population is mainly composed of youth and children [17]. Most of the inhabitants of Kawangware live on less than a dollar a day. The level of unemployment is high, and most of the adults are self-employed entrepreneurs. The majority of the families in this area is not able to afford more than a single meal a day, thus exacerbating their level of malnutrition [6]. Most of the children are also malnourished, particularly due to lack of proteins in their daily diet [24]. There are fewer government clinics and hospitals in Kawangware, and residents mainly rely on the healthcare services offered by non-governmental organizations such as AMREF, Medecins Sans Frontiers, and other good-will providers. Access to clean water is also a major problem among the slum dwellers. Overall, most of the families in Kawangware are subject to high poverty levels, and they struggle to fight disease and to support their families [25]. The poverty level among these slum dwellers classifies them as BoP consumers [16].

### 2.2. Sampling Procedure

Multistage sampling was employed to select the sample for this study as shown in Figure 2. The areas of interest were chosen purposely due to the presence of the BoP consumers. Systematic sampling was then used to choose respondents for the study. The study focused on the entire household, women of reproductive age (15–49 years), and children below five years of age (6–59 months).

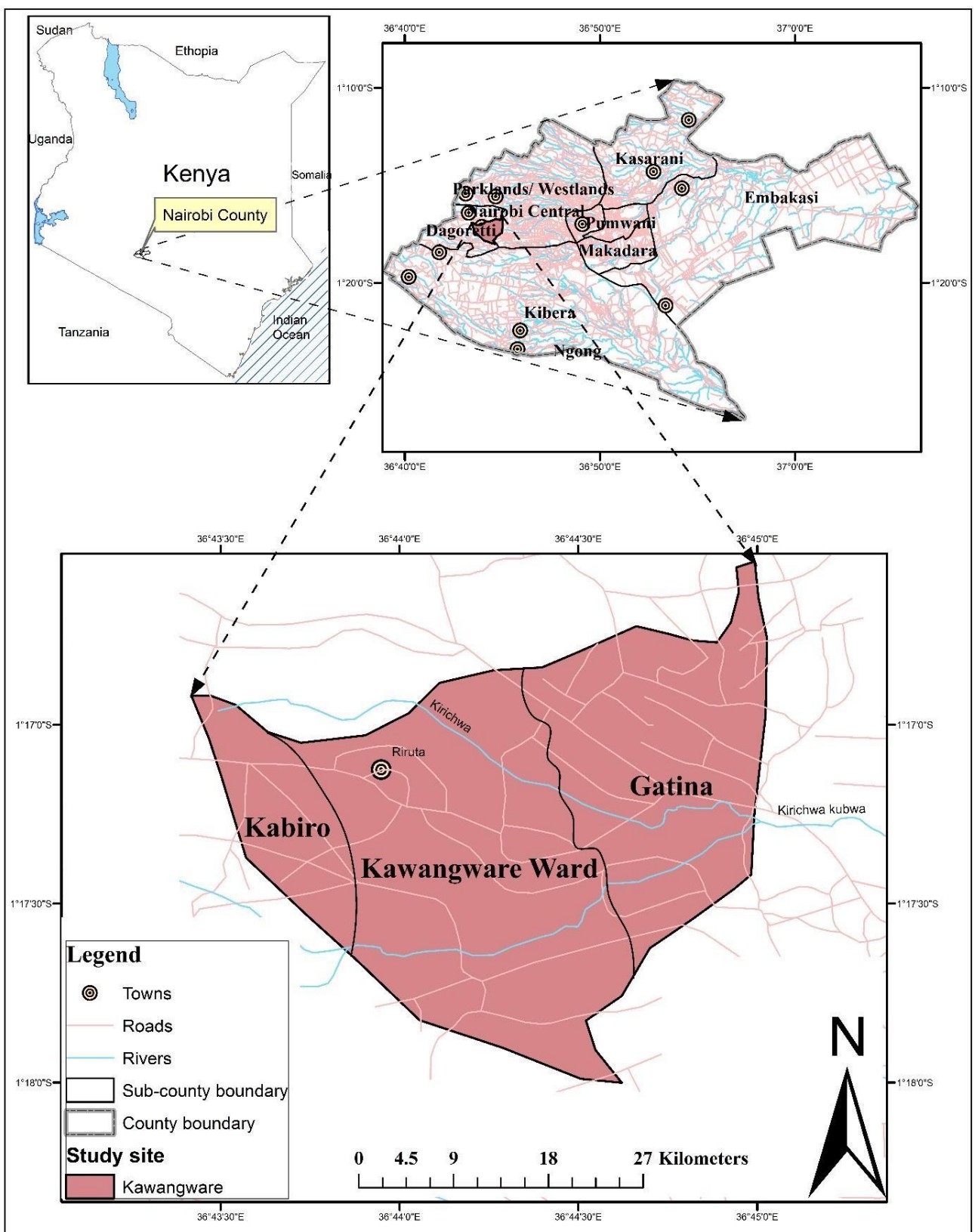

**Figure 1.** Map of the study area as drawn by the authors.

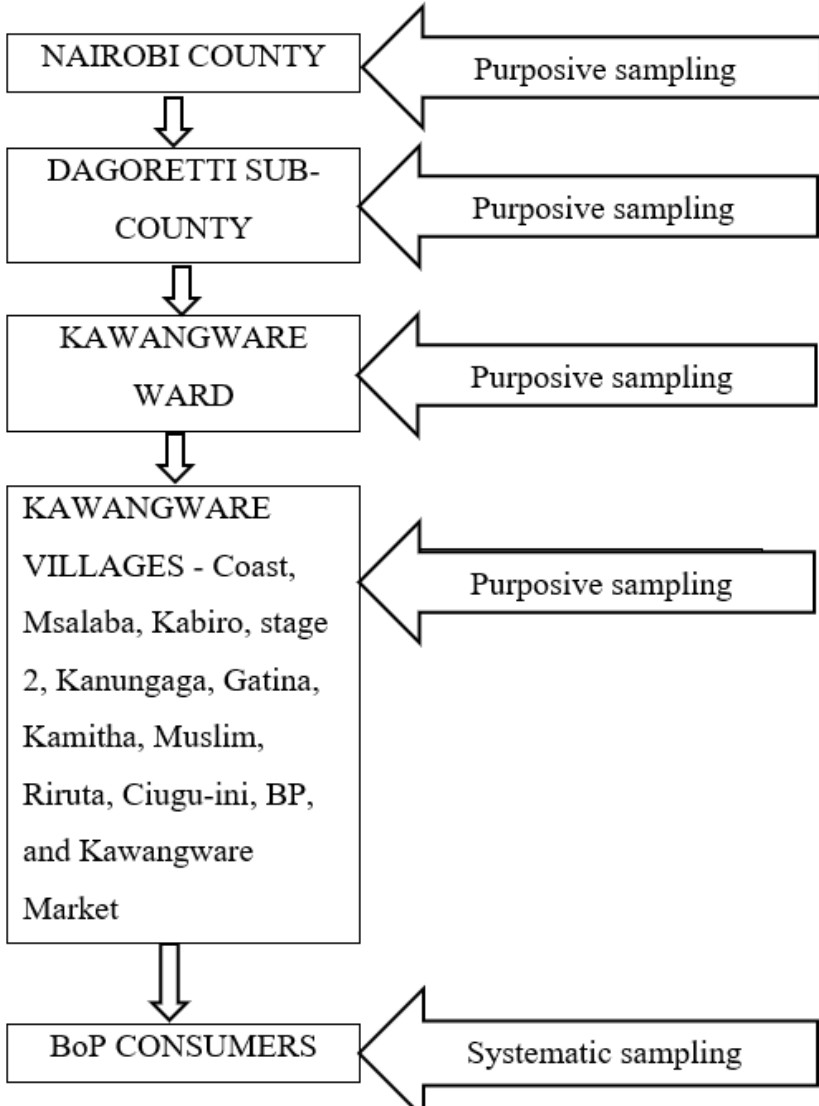

**Figure 2.** Multistage sampling procedure as illustrated by the authors.

The sample population was drawn from the household units of BoP consumers in Kawangware, [26] and a formula was used to determine the sample size since the population of the targeted study area was greater than 10,000.

The formula is presented below:

$$n = \frac{z^2 pq}{d^2}$$

where n is the desired sample size, z is the significance level, p is the prevalence level of malnutrition in slum areas [27], q = 1 − p, d is the degree of accuracy n = $(1.96^2 \times 0.4 \times 0.6) \div (0.05^2)$, and n = 603.

This study therefore collected data from a sample of 603 respondents representing the target population in Kawangware, Nairobi County, using a cross-sectional survey. A structured questionnaire was designed by the authors so as to capture all the study variables.

Systematic sampling was used to select the BoP consumers. The first respondent was selected randomly from the available household units, and the subsequent respondents

were selected by taking every 77th item, where 77 referred to the sampling interval. This process was repeated until the targeted sample size was achieved:

$$k = \frac{N}{n}$$

$$k = \frac{46,651}{603}$$

$$k = 77$$

### 2.3. Econometric Model

**Definition of outlets:**

*Supermarkets*—A self-service store that offers a large variety of foods, household products, and beverages. These foods can be highly processed, refrigerated, and frozen [9].

*Mom-and-pop shops*—These are individual/family-owned retail shops found in fixed locations, selling merchandise over the counter in small packaging, with a possibility of issuing credit to customers. They offer a moderate variety of brands and foods, processed staples, and some refrigerated foods [9].

*Kiosks*—This is a traditional retail outlet that is characterized by the sale of merchandise over the counter, offering a limited variety of brands, unprocessed staples, and fresh vegetables and fruits. Other features include offering merchandise in small packaging, giving credit to customers, and individual ownership of the respective outlet [9].

*Cereal stores*—This is a retail outlet that is characterized by the sale of a wide variety of cereals in both a processed and unprocessed state. Some of these stores operate in self-service mode, while others do not. The stores allow customers to choose various cereals which they feel suit their nutritional needs.

*Village health teams*—These are small groups that have been organized by the ministry of health at the village level to help in empowering communities to participate in making decisions which affect their health, mobilize communities for health programs, and enhance the delivery of health services at the household level. These teams act as a good means of increasing access to nutritious foods since they advise households on available nutritious foods and where they can get them.

This study assumed that BoP consumers purchase the nutrient-dense porridge flour from several outlets, such as supermarkets, mom-and-pop shops, kiosks, and cereal stores. These consumers can choose to purchase from one or more of these outlets. The most appropriate model was used to measure this objective, which is the Multivariate Probit (MV-Probit) model. This model aided in ascertaining the probability of a BoP consumer choosing one or more outlets over the others. The MV-Probit model was used to jointly estimate several correlated binary outcomes. The model is written as follows:

$$Y_{im} = \beta_{im} X_{im} + \varepsilon_{im} \tag{1}$$

where $Y_{im}$ $(m = 1 \ldots M)$ is the dependent variable of the market outlet chosen by the ith BoP consumer. These dependent variables are polychotomous variables that show whether porridge sales are made via the marketing outlet chosen by the consumer. These outlets were aggregated into five groups: supermarkets, mom-and-pop shops, kiosks, cereal stores, and village health teams. Each consumer could use 1, more, or none of these outlets. The independent variables, as described in Table 1, are (i) distance to outlet $(X_1)$, (ii) gender of the household head $(X_2)$, (iii) household income $(X_3)$, (iv) ready availability $(X_4)$, (v) consumer credit $(X_5)$, (vi) service mode used $(X_6)$, (vii) convenient operating hours $(X_7)$, and (xiii) level of hygiene $(X_8)$. The service mode used indicates whether the customer was served over the counter or on self-service mode, while consumer credit indicates whether the consumer could be sold goods on credit or cash terms only.

The most effective marketing outlet was ascertained by identifying the outlet where most pieces of the nutrient-dense porridge flour had been purchased in the last month.

The explanatory variables mentioned in the previous paragraph were used to explain the factors that determined the selection of the various marketing outlets. It was expected that the education level of the head of the household, household income, and the outlet's level of hygiene would have a positive impact on the use of supermarkets, while distance, consumer credit, and convenient hours of operation would have a positive impact on the use of kiosks and mom-and-pop shops. On the other hand, sellers' knowledge of the products being sold was expected to have a positive impact on the use of cereal stores and village health teams.

**Table 1.** Summary of the variables used in the MV-Probit model.

| Variable | Abbreviation | Unit of Measurement | Expected Sign |
|---|---|---|---|
| Distance to outlets | DSTNC | Meters | − |
| Household head sex Household income | HHSEX HHINCM | 1 = Male, 0 = Female KES | −<br>+ |
| Ready availability | AVLBLTY | 1 = Yes<br>0 = No | + |
| Consumer credit | CRDTPUR | 1 = Yes<br>0 = No | + |
| Service mode | SSMDE | 1 = Yes<br>0 = No | + |
| Convenient operating hours | CNVNCE | 1 = yes<br>0 = No | + |
| Good hygiene | HYGN | 1 = Yes<br>0 = No | + |

Notes: Analysis of MV-Probit model [28]. **Definition of certain variables in Table 1:** *Consumer credit:* Whether the outlet can offer goods to the consumer on credit. *Ready availability:* The nutrient-dense porridge flour is always stocked; hence, the customer can purchase it at every visit to the outlet. *Convenient operating hours:* Having flexibility in the hours of operation, allowing customers to access the outlet at any time of the day, including late in the evening. *Service mode:* Whether the outlet operates on a self-service basis or not. *Seller's knowledge of the products sold:* Whether the outlet operators are aware of the quality of the food products they offering and how those products will benefit the customer. *Outlet hygiene:* Whether or not the outlet operates in a clean space that guarantees food safety.

$X_{im}$ is a $1 \times k$ vector of independent variable(s) that influenced the decisions about the choice of marketing outlet, and $\beta_{im}$ is a $k \times 1$ vector of unknown parameters that were to be estimated, where $m = 1, \ldots, k$, with $m$ being the marketing outlets under consideration. Moreover, $\varepsilon_{im}$ and $m = 1 \ldots, m$ are the error terms that were distributed as multivariate normal, with each having a zero mean and variance–covariance matrix, $V$, where $V$ has a value of one on the correlations and leading diagonal.

The equations mentioned above are a set of equations shown below:

$$Y_{1i} = \beta_1 X_{1i} + \varepsilon_{1i} \tag{2}$$

$$Y_{2i} = \beta_2 X_{2i} + \varepsilon_{2i} \tag{3}$$

$$Y_{3i} = \beta_3 X_{3i} + \varepsilon_{3i} \tag{4}$$

$$Y_{4i} = \beta_4 X_{4i} + \varepsilon_{4i} \tag{5}$$

$$Y_{5i} = \beta_5 X_{5i} + \varepsilon_{5i} \tag{6}$$

The latent dependent variables are observed through a decision to purchase from the outlet or not, such that we obtain the following:

$$y_{mi} = \begin{cases} 1 \\ 0 \end{cases} if\ y_m^* > 0\ \text{or if} = 1,\ m = 1, 2, 3, 4, 5$$

There were only 10 possible combinations of choosing or not choosing the 5 outlets. The probability of having all the 5 outlets being chosen by a single consumer, "I", is given as follows:

$$\Pr\left(Y_{1i}=1,\ Y_{2i}=1,\ Y_{3i}=1,\ Y_{4i}=1,\ Y_{5i}=1\right)=\left(pr(\varepsilon_{1i}\leq\beta_1'X_{1i},\varepsilon_{2i}\leq\beta_2'X_{2i},\varepsilon_{3i}\leq\beta_{3i}'X_{3i}\ \leq\beta_{4i}'X_{4i}\ \leq\beta_{5i}'X_{5i}\right)$$

This system of equations is estimated jointly by using the maximum likelihood method. This estimation is conducted by using user-written STATA Mv-Probit methodology (Cappellari and Jenkins, 2003), which employs the smooth recursive conditioning simulator of GHK (Gewek–Hajivassiliour–Keane) to evaluate the multivariate normal distribution [29].

## 3. Results and Discussions

### 3.1. The Most Effective Outlet for the Supply of Nutrient-Dense Porridge Flour to BoP Consumers

Descriptive statistics were used to ascertain the most effective outlet through which to supply the nutrient-dense porridge to the base-of-the-pyramid consumers. The outlets that were used to supply the nutrient-dense porridge were mom-and-pop shops, kiosks, supermarkets, cereal stores, and village health teams. From the results shown in Table 2, the majority of the BoP consumers preferred to make their porridge-flour purchases from supermarkets (51.08%). This, therefore, qualifies supermarkets to be the most-effective outlet for the supply of nutrient-dense porridge flour to the BoP consumers. The second most-effective outlet was the use of cereal stores (25.54%), followed by kiosks (19.40%). Mom-and-pop shops (1.82%) and village health teams (2.16%) were the least-used outlets in the purchase of the nutrient-dense porridge flour.

**Table 2.** Outlets used by BoP consumers to avail themselves of nutrient-dense porridge flour.

| Source of the Flour | Freq. | Per Cent | Cum. |
|---|---|---|---|
| Mom-and-pop shop | 11 | 1.82 | 1.82 |
| Kiosk | 117 | 19.40 | 21.23 |
| Supermarket | 308 | 51.08 | 72.31 |
| Vendor | 154 | 25.54 | 97.84 |
| Village health team/counselor | 13 | 2.16 | 100.00 |
| Total | 603 | 100.00 | |

An MV-Probit model was used to ascertain the factors influencing the respective outlets chosen by the BoP consumers. Table 3 shows the pairwise correlation coefficient between the error terms of the five equations of outlets' usage. All ten pairs of the estimated correlation coefficients were statistically significant from zero, therefore implying that there was a strong interdependence among the five outlet types in usage for access to nutrient-dense porridge [30].

**Table 3.** Correlation coefficients for MV-Probit model's regression equations.

| | Supermarket | Kiosk | Mom-and-Pop Shop | VHT | Vendor |
|---|---|---|---|---|---|
| Supermarket | 1.000 | | | | |
| Kiosk | −0.439 *** | 1.000 | | | |
| Mom-and-pop shop | 0.208 *** | −0.436 *** | 1.000 | | |
| Village health team | −0.124 *** | −0.023 *** | −0.221 *** | 1.000 | |
| Vendor | 0.050 *** | −0.030 *** | 0.070 *** | −0.197 *** | 1.000 |

Notes: *** Represents significant level at 1%.

### 3.2. Factors Influencing the Choice of Outlets among the BoP Consumers

Table 4 shows the MV-Probit model results that revealed the significant variables that influenced the usage of the identified outlets among the BoP consumers. The Wald test $[x^2(45)=175.11,\ p<0.0000]$ implied that the data that were used were fit for the MV-Probit model, and the likelihood ratio $[x^2(10)=158.92,\ p<0.0000]$ of the independence

of multiple usage of various outlets was strongly rejected. This shows that the multiple usage of different outlets among the BoP consumers is not mutually exclusive.

**Table 4.** Multivariate Probit's marginal-effects results for the factors influencing the choice of outlets among the BoP consumers.

| Outlets | Supermarket | | Kiosk | | Vendor | | Mom-and-Pop Shop | | Village Health Team | |
|---|---|---|---|---|---|---|---|---|---|---|
| Variable | Marginal Effects | SE | Marginal Effects | SE | Marginal Effects | SE | Marginal Effects | SE | Marginal Effects | SE |
| Distance to outlets | −0.0004 ** | 0.0001 | 0.0001 | 0.0001 | 0.0001 | 0.00005 | −0.0001 | 0.0001 | −0.0001 | 0.0001 |
| Household head's sex | 0.5991 | 0.4689 | 0.1384 | 0.1093 | −0.1383 | 0.1136 | 0.0318 | 0.1035 | −0.0890 | 0.1032 |
| Household income | −0.0001 | 0.0001 | 0.0001 | 0.0001 | 0.0001 | 0.0000 | −0.0001 | 0.0000 | 0.0001 | 0.0000 |
| Ready availability | −0.2428 | 0.5300 | 0.1247 | 0.1766 | 0.0777 | 0.1856 | 0.2856 | 0.1756 | 0.1119 | 0.1722 |
| Can purchase on credit | −1.4025 ** | 0.6164 | −0.5677 | 0.4214 | 0.7177 * | 0.4138 | 0.2541 | 0.3884 | 0.4443 | 0.4150 |
| Self-service | 1.6317 ** | 0.8466 | −0.8641 * | 0.4537 | 1.1697 ** | 0.4640 | 0.2048 | 0.4281 | 0.7918 * | 0.4555 |
| Convenient operation hours | −8.1617 | 175.8710 | 0.5327 *** | 0.2164 | −0.2368 | 0.2278 | 0.2539 | 0.2073 | −0.1444 | 0.2022 |
| Hygiene level | −8.5036 | 202.1719 | 0.8887 *** | 0.1690 | 0.0203 | 0.1775 | −0.6794 *** | 0.1680 | 0.3594 ** | 0.1626 |

Note: *, **, and *** = significant at 10%, 5%, and 1% level, respectively. SE is Standard error.

The distance to the outlet from the household location had a negative effect on the use of supermarkets, with an additional unit increase in distance reducing the probability of supermarket usage by 0.0004 percentage points. These results indicate that an additional distance to an outlet is associated with lower probability of the household choosing to use the supermarket as a source of nutrient-dense porridge. This coefficient was statistically significant at a 5% level of significance. Distance to the nearest outlet determines the ease in the accessibility of commodities by consumers. Most consumers prefer to carry their shopping from retail outlets that are located close to their residential areas. Consumers also tend to establish close ties with those running the retail outlets that are near to their households, and this makes it very difficult for them to switch to those that are located slightly further away, unless absolutely necessary. Purchasing from an outlet that is close to the household is also very convenient, as it reduces travel across long distances [31]. The close proximity of outlets offering nutritious foods to households is one of the key factors which play a great role in ensuring that food and nutritional security is enhanced in the society [32].

Household income had a positive impact on the use of cereal stores as the preferred choice of outlet. This implies that one unit increase in the income level of the household head increased the probability of purchasing the nutrient-dense porridge from cereal stores by 0.0001 percentage points. This variable was statistically significant at the 5% level of significance. A household tends to be more cautious about the health benefits of what the members consume as income increases. Unlike the other outlets being considered, cereal stores have been known to supply various types of products that can be blended to prepare a porridge flour that is highly nutritious and meets the desired health benefits. This is the reason why the household heads with higher incomes increasingly preferred to purchase the nutrient-dense porridge flour from the cereal stores. Household heads whose income is low have no other option than to consume what is available at the market. This is because their incomes cannot allow them to choose a variety of products in the market that will enable them to end up consuming a healthy diet. They can only afford to purchase food to quell their hunger and that provides them with enough energy to get by, regardless of whether the food offers health benefits. BoP consumers will continue to face the issue of food insecurity as we move into the future, unless adequate interventions are introduced, since a majority of them struggle to afford any type of food, let alone what is nutritious. Rising food costs, together with other shocks, such as economic crises, foods, and drought, have major impacts on food and nutrition security since they push vulnerable households, such as those of the BoP consumers, further into poverty, therefore weakening their ability to have access to sufficient food [32].

The ability to purchase on credit had a positive impact on cereal stores and a negative impact on supermarkets. These variables were found to be statistically different at a 10% significance level for the cereal stores and a 5% significance level for the supermarkets. Cereal stores that could sell their porridge on credit had a higher probability of attracting the

BoP consumers, unlike the supermarkets. Providing access to credit increased the number of nutrient-dense porridge units that could be accessed through cereal stores by 0.7177 percentage points. The negative impact that credit access had on the use of supermarkets was expected, considering that it is very rare to find supermarkets offering goods to consumers on credit. Supermarkets would rather supply the retailers on credit rather than the consumers for many reasons, including the logistics involved in collecting cash from the customers who owe money to the supermarket. Cereal stores attract consumers who would shop at the supermarket, if given the access to credit. The majority of individuals who run the cereal stores usually have close relationships with their customers, and this facilitates the process of selling goods on credit. It is also worth mentioning that the majority of the BoP consumers do not always have access to sufficient cash, and during such times, they need to find alternative ways of obtaining food. Buying food on credit is the most common alternative available to them during such situations. Supplying the nutrient-dense porridge flour through cereal stores is, therefore, one of the ways of dealing with food insecurity in the slums since it almost always guarantees the BoP consumers a constant supply of food, as long as they show a good credibility in settling their debts. Sale on credit increases access to food in the slums, and the use of this alternative is, therefore, helpful in reducing the high prevalence of food insecurity among BoP consumers [33]. Food security will only exist when all individuals, at all times, have both economic and physical access to sufficient, nutritious, and safe food which meets their respective dietary needs and preferences for a healthy and active life [34]. Thus, access to credit plays a very vital role toward increased food accessibility, therefore enhancing food security.

The ability to purchase porridge in self-service outlets had a positive impact on the use of supermarkets and cereal stores and a negative impact on the use of kiosks as a source of accessing the nutrient-dense porridge flour. This implies that supermarkets and cereal stores selling their porridge on a self-service basis had higher probabilities of attracting the BoP consumers compared to the kiosks. This was expected in the case of kiosks because they are not designed to allow users to serve themselves. The self-service strategy increased the units of the nutrient-dense porridge flour purchased through the supermarkets by 1.6317 percentage points and by 1.1697 for the cereal stores. These coefficients were statistically significant at 5% significance level. Self-service stores are convenient and attractive to most consumers since they allows the customers to easily compare different commodities before purchasing them. A majority of the BoP consumers compare the commodities in terms of price before they start comparing other aspects/elements, such as quality, brand, and packaging, among others. This makes the price element a key indicator. Thus, it is imperative to ensure that the prices of nutrient-dense food products are competitive and affordable, so that they can easily penetrate the informal urban markets. This approach will contribute to solving the issue of food insecurity, which is currently a key challenge experienced by dwellers in these settlements. Food affordability in the slums is an important factor toward solving food insecurity since it enhances the access to various food commodities available on the market [35]. It increases the purchasing power of the BoP consumers, therefore enabling them to access food that is adequate in both quality and quantity and thus boosting their food and nutritional security [34].

Convenient hours of operation had a positive impact on kiosks, implying that kiosks operating during hours that were convenient to the BoP consumers had a higher likelihood of attracting these consumers for the sale of porridge. From our survey results, we determined that kiosks that operated during convenient operating hours were highly likely to increase sales by 0.5327 percentage points. This variable was found to be statistically different at a 1% significance level. Economies in the urban areas operate both during the daytime and at night, making it possible for outlets to operate 24 h per day so that they can always cater to the needs of their clients. The majority of the individuals who run the kiosk businesses in the slums usually live around their business premises, thus allowing them the flexibility to operate their businesses during odd hours, unlike those running the other types of outlets considered. This situation offers a competitive advantage to kiosks to some

extent. It is vital for companies to give special attention to kiosks when they are identifying outlets that can be used to supply commodities such as the nutrient-dense porridge flour to BoP consumers in the informal urban settlements. The use of such effective distribution outlets is one of the best strategies for more easily penetrating the market, while also enhancing the consumption of nutritious products among the BoP consumers. Directly supporting these types of outlets would also allow them to save on the resources that could be used to run certain marketing campaigns. Customers are always loyal to retail outlets that operate during the hours that best suit them. These results conform the findings of Reference [36], who found that outlets that operated during times that were more convenient to the target customers were more likely to attract more customers compared to those that did not consider this factor. With such convenience, BoP consumers are able to have guaranteed access to nutritious foods, thus contributing to improved food security.

The hygiene levels of the outlets selling nutrient-dense porridge flour also played a key role in influencing BoP consumers' levels of purchase. Our survey results showed that high levels of hygiene increased the consumption level of the nutrient-dense porridge flour by 0.889 and 0.359 percentage points in the use of kiosks and village health teams, respectively. On the other hand, poor levels of hygiene in the use of the mom-and-pop shops were more likely to reduce the consumption level of the nutrient-dense porridge flour by 67.94%. The creation of a good retail environment helps to increase the length of time the shopper will spend in the outlet and generates loyalty with the established brand. It also encourages people to be repeated customers. After pricing, hygiene is one of the main factors that plays a vital role and aids in the creation of such a good environment, particularly when customers are shopping for food commodities. This is because they believe that a clean outlet will supply goods whose quality has not been compromised by factors that lead to poor hygiene. A hygienic environment will enhance the shopping experience among customers and the overall impression that they have toward the brand and the shop. This implies that they will always shop at the outlets with good hygiene, as demonstrated by the results of our study. A superior experience is a unique and valued asset for any type of business, especially considering that competition is high, and every business wants to maintain its customer base and attract more of them [37].

## 4. Conclusions

Although the consumption of nutritious foods is still low among BoP consumers, this study has shown that this trend can be reversed, thus helping to fight food and nutrition insecurity. Supermarkets, cereal stores, and kiosks play a very vital role in ensuring that access to nutrient-dense foods is enhanced in the informal urban settlements. Mom-and-pop shops and village health teams were the least used outlets in the purchase of the nutrient-dense porridge flour. Despite the village health teams being the least preferred BoP consumer choice, so far, they have played an instrumental role in increasing awareness on the importance of consuming nutritious products. Ensuring that the right outlets are stocked with sufficient quantities of nutritious products will aid in addressing food- and nutrition-insecurity problems in the informal urban settlements, since they will enhance the accessibility of these products. This study makes a significant contribution to the body of knowledge in understanding how marketing approaches can and should be used to address the issue of food and nutrition insecurity in informal urban settlements. Further research should also be conducted with regard to ascertaining the impact of different institutions on improving food security in the informal urban settlements of Kenya and globally.

**Author Contributions:** Conceptualization, K.K.K. and C.G.K.C.; methodology, K.K.K.; software, K.K.K.; validation, K.K.K., C.G.K.C. and H.B.; formal analysis, K.K.K.; investigation, K.K.K.; resources, C.G.K.C.; data curation, K.K.K.; writing—original draft preparation, K.K.K.; writing—review and editing, K.K.K., C.G.K.C. and H.B.; visualization, K.K.K.; supervision, C.G.K.C. and H.B.; project administration, C.G.K.C.; funding acquisition, C.G.K.C. All authors have read and agreed to the published version of the manuscript.

**Funding:** This research was financially supported by the German Federal Ministry for Economic Cooperation and Development (BMZ), Germany, through the project number 15.7860.8-001.00, "Making Value Chains Work for Food and Nutrition Security in East Africa". Additional funding was provided by the flagship program "Food Systems for Healthier Diets", under the CGIAR Research Program on Agriculture for Nutrition and Health (A4NH). The donors were not involved in the research in any way. The project was led by the Alliance of Bioversity International and the International Centre for Tropical Agriculture (CIAT) and implemented in partnership with other local and international organizations, including Egerton University.

**Institutional Review Board Statement:** The research questionnaire was developed in collaboration with scientists from the Kenya Agricultural Livestock and Research Organization (KALRO) in Kenya. In addition, ethical clearance was in accordance with the Kenya Agriculture and Livestock Research Act, No. 17 of 2013, Article 5, Section 2d; and the Science, Technology, and Innovation Act, No. 28 of 2013, Part IV, Article 12, Section 2.

**Informed Consent Statement:** Informed consent was obtained from all subjects involved in the study prior to start of the interviews.

**Data Availability Statement:** The data presented in this study are available upon request from the corresponding authors.

**Acknowledgments:** The authors of this paper thank the Kenya Agricultural and Livestock Research Organization (KALRO) for their cooperation and support during the fieldwork. This article is part of an MSC thesis supported by Egerton University. We thank Olga Spellman (Alliance of Bioversity International and CIAT) for English and copy editing of this manuscript.

**Conflicts of Interest:** The authors declare no conflict of interest.

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
