# Peer review of "Which Food Outlets Are Important for Nutrient-Dense-Porridge-Flour Access by the Base-of-the-Pyramid Consumers? Evidence from the Informal Kenyan Settlements"

_sustainability, doi:10.3390/su141912264_

Round 1
Reviewer 1 Report
Dear Authors,
I enjoyed reading your manuscript, I think your research topic is very important. You did a good job with sampling, it looks like the sampling procedure was rigorous and well designed, and it’s likely that the sample is representative of the sampled population.
On the other hand, literature review seems minimal, and perhaps it can be expanded. For example, were there any studies that looked at preferences of the BoP or similar consumer group regarding nutrient-dense foods, or that evaluated efficiency of different marketing approaches when selling food to socially disadvantaged consumer groups in the past? Also, it would be helpful to mention some studies (if available) that looked at issues related to access to nutrient-dense food among socially disadvantaged populations in general.
More importantly, I am not sure about the model used in the analysis and interpretation of the results. Based on my knowledge, probit model is used when the dependent variable is binary, which might or might not be the case in this study. The dependent variable was not clearly defined, and so I am not sure whether the model is appropriate, and whether the results were interpreted correctly. Other parts of the model were not described clearly either.
I also missed a mention of the study limitations at the end of the manuscript. Finally, the manuscript needs technical editing, in addition to grammar and spelling check.
Specific comments:
Lines 19-20: “The use of right marketing strategies leads to increase in uptake and consumption of nutritious products in the urban informal settlements” – is this something that was found in this study, did the results of this study lead to this conclusion? I do not think that is the case, and it should be made clear that this is not a finding of this study, but rather a possibility or hypothesis that could be investigated.
Lines 41-43: Who categorized these consumers as the base of the pyramid? Is there a study that can be cited?
Line 52: What exactly is meant by “diversification” of diets here? I understood that it meant that the diet should consist of a variety of foods. But later, in lines 63-65, it looks like the diversification meant here is to enrich a specific food item with more nutrients? Also, is there a study that discussed diversification as one of the approaches to reduce malnutrition, which could be cited, and what are some other approaches?
Line 71: What is the difference between accessibility and availability?
Line 97: “… 15 Kilometres to the West of Nairobi County.” – this is not clear.
Lines 111-112: “The poverty level among these slum dwellers makes them be classified as BoP consumers.” – is there any citation for that?
Line 114: Multistage sampling.
Line 128: Z or z? There is Z in the formula, not z.
Line 149: “otherwise” – meaning, not purchasing any porridge at all or something else?
Lines 151-152: What is meant by “others” here?
Line 152: Based on my knowledge, probit models are used when the dependent variable is binary, i.e., when an event (such as purchase of a product) happens or not, but it is not clear what the dependent variable is in this study. Was it a binary outcome or was it the count of the porridge flour units consumed? If the second, were the units of the same size? If yes, what was the size? If not, explain why is it appropriate to use that in the analysis? I believe it is important that the unit of measurement is the same across all respondents.
Also, it is not clearly explained by authors why multivariate probit is applicable in this study, and what is the theory behind using this model. If the dependent variable is indeed not binary but instead it is a count of porridge flour units, authors could use ordered probit model instead.
Line 155: What do the subscripts “m” and “i” represent?
Lines 156-157: “These dependent variables are polychotomous variables which show whether porridge sales are made via the most relevant marketing outlet” – it is not clear what the dependent variable is… Is it the number of porridge flour items purchased in a given store, or is it a binary variable showing whether the purchase of flour was made in a given store?
Lines 158-159: Did they also have an option to choose none of these outlets?
Lines 159-163: Please mention explicitly which terms in the probit equation (i.e., vector X) correspond to these independent variables. And how are coefficients P(im) interpreted, with respect to the dependent variable? I also see that X and P are discussed in lines 164-173, and so these two paragraphs could probably be reorganized and edited for clarity.
Line 159-160: Is quantity demanded an independent variable or was it the dependent variable?
Line 160: Was it a dummy variable for the household head, i.e. 1 if household head took the survey and 0 if someone else than the household head took the survey? Or was it the gender of the household head? Please clarify.
Lines 166-170: These are enumerated in the previous paragraph already, perhaps it's better to refer to them shortly as explanatory variables. Also, not clear what some of these mean and is there a theory why these were chosen? Any expectations regarding the results?
Line 173: There should be m=1,…,k.
Line 175: “V has values of one on the correlations and leading diagonal” – double check the wording.
Lines 177-179: What do the subscripts 1,2, and 3 represent? Are these the marketing channels? If yes, why not 4?
Line 180: “to adopt or not to adopt” – I would use different words here, i.e., purchase from the outlet or not.
Line 182: What are the subscripts “i” and “j”? Should there be “m” instead of “j,” and also the order should be “m” first and “i” second, in line with the notation in the previous equations? And should there be “m” instead of “k”?
Line 184: I thought there were 4 outlets, not 3?
Lines 186-188: Are these three equations correct/necessary? It appears that they express the same thing. Also, I believe all equations in the manuscript should be numbered.
Lines 195-198: Move the table with notes after it is mentioned in the text. Also, the table needs to be mentioned in the text, which it isn’t currently. Also, double check that the rows show in the correct order, it looks like some rows were moved around and some items are missing (e.g., no gender in the variable column). Also, if the expected sign is +/- for all variables, why report it?
Lines 201-203: Now, there are 5 outlets, while only 3 or 4 were mentioned before. Please fix so that the number of outlets is consistent across the manuscript. Also, can authors provide a definition/description of each studied outlet?
Line 205: I do not think it can be concluded that it is the most effective outlet. Effective from what perspective? Is it perhaps most accessible?
Line 213: There are more than 5 pairs, I counted 10 in table 3.
Lines 213-215: So, what do these strong correlations mean? It looks like they provide justification for why multivariate probit is appropriate for the analysis. Which also makes sense, since if consumers purchase more from one outlet, then they will have less need to purchase from other outlets, given their demand and resources are limited. In other words, it appears that the purchase of flour from a given outlet does not occur independently of the other outlets, thus provides some evidence that multivariate probit is indeed appropriate. Authors might want to mention that in the manuscript.
Table 4: What does “Ready availability” variable mean? Also, it looks like there are 12 variables in table 1, but only 9 are reported in table 4 – please double check and/or explain why some are missing in table 4. Why were the three variables dropped?
Line numbers were not available in the last three pages of the manuscript text, so I will copy and paste the sentences with my comments:
- “This shows that the multiple usage of different outlets among the BoP consumers is not mutually exclusive.” – also, the demand is limited, so if consumers purchase more from one outlet, they will automatically reduce their purchases from other outlets.
- “… reducing the consumption of nutrient-dense porridge flour by 0.0004 units.” – from the discussion of results, it is not clear whether the dependent variable was binary (purchase/no purchase from the outlet), which it should be if a probit model is used, or whether it was a count of units of flour purchased, which would imply that another model (perhaps ordered probit) should be used.
- “These results indicate that an additional distance to an outlet is associated with low probability of the household choosing..” – I would say “lower” probability instead of “low”
- “This variable was statistically different at 5% level of significance” – not variable, but coefficient, and it was statistically significant, not different, at 5% level. Also, later in text it should be statistically significant and not statistically different.
- “Household head income had a positive impact on the use of vendors as the preferred outlet choice.” – looking at the results in table 4, it is not shown as statistically significant there – please double check and correct as needed
- “This is the reason as to why the household heads whose incomes were increasing” – I would say “household heads with higher incomes,” the way it is written in the manuscript suggests that respondents’ incomes were increased during the survey and the effect of the increase was measured
- “An additional access to credit increased the number of nutrient dense porridge units that could be accessed through vendors by 0.7177” – it is not clear what is meant by “additional access to credit”? I see that the variable was measured as yes or no, I suppose depending on whether the credit is available at the outlet? Then, I would not say “additional access to credit has increased the number,” but rather “providing access to credit increased the number…”
- “The negative impact that credit access had on the use of supermarkets was expected considering that it is very rare to find supermarkets offering goods directly to consumers on credit.” – could the reason also be that the access to credit at vendors, which increases purchases at vendors, reduces purchases at supermarkets? I am thinking that the sum of purchases across all outlets would be relatively fixed, and only the share of each outlet may change, when credit is provided at some outlets (such as vendors), which would then mean that the share of others (such as supermarkets) would decrease. Can we also say that vendors attract consumers who would shop at the supermarket, if given the access to credit?
- “They rather supply them to retailers on credit” – it’s not clear what authors mean here.
- “Ability to purchase porridge on self-service had a positive impact on the use of supermarkets and vendors” – also village health teams, based on results in table 4. And how common is this self-service, and how does it work usually? Does it mean that they check out by themselves, or can they pick the product by themselves and bring it to the register? Or what does it look like when the self-service is not available, is it just selling over the counter? It might be helpful to describe a bit what these terms mean in case there are differences compared to other countries.
- “These variables were statistically different at 5% significance level” – not these variables, but coefficients, and statistically significant, not different.
- “Convenient hours of operation had a positive impact on kiosks” – also here, was it defined what is meant as convenient hours of operation to respondents, or were they left to decide for themselves what they consider convenient and no definition/information about what is convenient is available? I am wondering if there is some reference point as to what convenient operation hours are.
- “hours were highly likely to increase their sales by 0.5327 units” – it’s more appropriate to say that the sales increase by 0.5327 units.
- “the hygiene level” – was that defined to respondents or were they left to interpret it themselves?
- “increased the consumption of level of the nutrient-dense porridge flour by 88.87% and 35.94%” – is this the correct way to interpret the coefficients? If yes, then the interpretation of the coefficients for previously discussed variables should be updated. Otherwise, the interpretation for this variable should be corrected (i.e., hygiene level increased consumption by 0.889 units for kiosks, etc.)
Reviewer 2 Report
The manuscript deals with an important and very topic problem. The text structure is correct. Appropriate methods were also used. The selection of respondents is appropriate. In the introduction, the authors justified the need for research and presented the research problem. However, it is necessary to indicate the purpose of the research ... .. Of course, the purpose of the research has been formulated, but from the formal point of view (in my opinion) the following phrase should be used: the purpose of the research was ....
The research results are interesting and can be put into practice, which adds value to this work. The manuscript is well prepared, but I have some comments. First, the information on the variables used should be completed. For example, who and how assessed Environment's hygiene? Please also justify why the "product price" variable was omitted in the analyzes? If the price in different stores is different, it can have a significant impact on purchasing decisions.
The Probit Model is an effective tool, but in the case of categorical variables, a better solution would be to use, for example, decision trees. The authors may use it in their future studies. Interesting results could be obtained from The Analytic Hierarchy Process (AHP).
Minor Notes:
- Table 1 should be included in chapter 2.3, not chapter 3.
- - Please adapt the citation method to the editorial requirements.
- Please correct over the literature review - too few references
The final evaluation of the manuscript will depend on the answer to the question why the authors omitted the "price" variable.
Reviewer 3 Report
The study deals with intervening factors in food choice that can aggravate food insecurity present in different world populations. The authors highlight the need to investigate these aspects in consumers at the base of the pyramid. The absence of similar data related to the target audience surveyed reveals the importance of the study. Furthermore, the results suggest the need to consider market strategies as part of actions to combat food insecurity. In addition to nutritional quality, the results indicate the role of environment with favorable conditions such as hygiene of the place, location, among other factors, on food sale.
However, some clarifications are necessary.
Line 58 – 66: “Majority of the porridge that these informal settlement residents consume are not diversified. For instance, most times the porridge is cereal-based, and it contains only one food item, such as maize, or millet or a combination of the two, which eliminates hunger but lacks important micronutrients. Addressing hunger alone is not sufficient! There is need to address micronutrient deficiency as well. Therefore, this would entail diversifying porridge flour to include other food items which will provide important micronutrients to consumers. Note that diversifying the porridge flour may imply making the flour more expensive so that most of the BoP consumers cannot be in a position of affording it.”
Comment: The information the authors provided is important and defines an important diet quality problem. Therefore, the authors must include the references of the study that evaluated the food consumption and/or nutritional status of this population that evidences the statement.
Line 84 - 86: “To help in addressing this challenge, the main objective of this study was to assess the consumption of nutrient-dense porridge flour by the base of the pyramid consumers in the informal settlements of Kawangware in Nairobi.”
Comment: It is unclear what nutrient-dense porridge flour is. I suggest that the authors be more specific about what this is about.
Line 90 – 93: “369 respondents were interviewed in this study. The study put emphasis not only on the entire households but also on women of the reproductive age (15-49 91 years) and children below five years (aged 6−59 months). The latter are the most vulnerable in terms of attaining a healthy and nutritious diet.”
Comment: It would be more appropriate for this snippet to be located in the topic “materials and methods”.
Conclusions: “Though consumption of nutritious foods is still low among BoP consumers, this study has shown that this trend can be reversed therefore helping in combating food nutrition security.”
Comment - Would not be: “…in fighting food nutrition Insecurity.”
Reviewer 4 Report
The manuscript "Effective outlets of availing nutrient dense foods to the Base of Pyramid consumers" is an effort to explore the factors influencing the purchase from effective outlet . to improve the quality of the article i have some major suggestions.
Authors should explain their research problem by focusing on "they want to access the consumption or the effective outlet from which the respondents purchased the flour"
Line 111-112: Although the author focused on Base of Pyramid Consumer in introduction, but how they classified the residents of Kawangware as BoP is missing. They should first explain more why they are BoP consumer in introduction section. Moreover, they should explain why they are important to assess their purchasing from which outlet.
Moreover, authors should add the figure of study area which should depict real picture of whatever they wrote in study area section.
Line 149: author should complete sentence “ utility greater than …..?
Line 159-162: authors should explain why they selected these independent variable specially why they incorporated “meeting with friends at the purchase outlet” into model? What is its importance in choosing the outlet for purchase….
Line 164-165: Why author ascertained the most effective outlet? have they used it to specify the model…. If yes then how? If no then why they ascertained it….? have they used it to specify the dependent variable of the model. At which criteria they supposed the which outlet is effective?
Table 1: how they incorporated the number of years of schooling in one variable as they used education as categorical variable.
Can meet with friend” variable needs to justify…?
How “Convenient operating hours” is specified. Did author know the Convenient hours of all the respondents …? Needs to explain.
The variable “Seller’s knowledge of products sold” is not easy to measure in form of yes/no. Author should justify it how they measured it?
Line 199-209: Again same confusion, how authors knew the most effective outlet for purchase of nutrient dense flour? If they assessed it, then they should explain how they differentiated which one is most effective according to the respondent. Have they asked them that it is an effective store and how many times they purchased flour from effective outlet.
Table 2: Author should focus on the result of the table. they have given the frequency which depicts that how many times respondent purchased the flour from which outlet in a specific time period which also based on the memory of the respondent. Therefore, they have no need to present the total values and how they can calculated the percentage based on the random responses.
Section 3.2: How can author say that, 1 unit increase in distance from outlet, the consumption of flour is reduced by 0.0004 units. While the dependent variable is not quantity of flour consumed from supermarket…
Why author extensively focused on the insignificant variable while they have no effect on purchasing from various outlets. Author should check your table 4 and also table 1. Variable are very different. Hypothesised variable were not used in actual model why?
Round 2
Reviewer 1 Report
Dear Authors,
the manuscript has been improved, but there are some more details that need to be added to the manuscript or clarified. Please see my comments below.
Line 20: I would not use citation in the abstract. Instead, you can say “According to past studies, the use of right marketing strategies 18 leads to increase in uptake and consumption of nutritious products in the urban informal settlements.”
Line 52: Please specify in the manuscript what you mean by the “diversification of diets” as relevant for this article.
Line 71: Please describe the difference between accessibility and availability here in the manuscript.
Lines 159-160: “… if this gives them a greater utility” – please specify, greater utility compared to what? No purchase?
Line 165: I can see that subscript “I” denotes customer (i-th customer), but what is subscript “m” in the equation? It looks like it might be used to denote marketing outlet, since in line 198, there is m=1,2,3,4,5, and 5 is the total number of marketing outlets. Please clarify.
Line 166: I see “m=1,…,k” here, while there are also “1xk” and “kx1” vectors mentioned in lines 185-186, and again “m=1,…,k” on line 187. It looks like “k” is used in different contexts (as the number of marketing outlets and number of independent variables) and it is confusing. I suggest that perhaps you change “m=1,..,k” to “m=1,…,M” where M is the total number of marketing outlets.
Line 168: Why saying “the most relevant” outlet here? I thought the analysis was done for every of the five studied outlets.
Line 170: Could they use none of these outlets?
Line 171: I counted only eight variables in Table 1, why are four missing? Please explain in the manuscript.
Line 172: “distance to the nearest marketing outlet” – was it measured considering just one of the four marketing outlets (the one of the four that was closest), or was the closest one identified for each of the four (or five?) marketing outlets?
Line 173: Please specify more “consumer credit” (access to it?) and “service mode used” in the manuscript.
Line 176-177: “The most effective marketing outlet was ascertained by identifying the outlet where most pieces of the nutrient-dense porridge flour had been purchased from in the last one month.” – this was done considering the entire sample or for each individual respondent? Also, if the information on the number of purchased flour packages per outlet was collected, please report that in the manuscript, perhaps in Table 1.
Line 184: “vendors and village health teams” – these were not mentioned in the manuscript until this point. Or, are these the “cereal shops” mentioned before when enumerating the outlets (see lines 157 and 169)? Please clarify it in the manuscript (use a consistent terminology).
Line 185: “??? is a 1 x k independent variable…” – isn’t this a vector of independent variables? Please specify in the manuscript as needed.
Lines 186-187: Again, what does subscript “m” stand for? I am guessing it is the marketing outlet, but please be specific here.
Line 197: There is “???” in the text, I assume by accident.
Line 198: “?? ?∗ >0” – this should be on the same line as y=1.
Lines 199-200: There are more than 10 probabilities for possible combinations of outlet choices. Or, did you mean to say that there were only 10 possible combinations observed in the sample? Please correct or specify, as needed.
Line 200: There should be “all the 5 outlets,” not 3.
Lines 202-203: Here, you use beta coefficients, but in the previous equations (lines 165, 191-195), the coefficients are denoted by P. Please choose one and correct as needed.
Line 210 (Table 1): Please move table 1 somewhere else in the text, it should not be right after the heading “Results and Discussion.” Perhaps move the table after the paragraph where it is discussed. Is the same sign expected for each marketing outlet? Looking at table 1, it looks like that is what you mean, but then hypotheses in lines 179-184 suggest that the results differ across the outlets. Also, why is there both + and – for the same variable? Finally, please add summary statistics for these variables (at minimum, sample averages).
Lines 214-222: Some of these variables sound subjective, were they defined somehow to the respondents? Please comment on that in the manuscript, whether yes or no. Also, it would be better to include the definitions in the text, where these variables are mentioned for the first time (i.e., where the econometric model is explained). Also, what does “ready availability” in table 1 mean?
Lines 244-252: These definitions are useful, but it would be better to incorporate them in the text. For example, include them in the section where the econometric model is described (that is where you mention these outlets for the first time). Also, do all supermarkets offer self-service only and do all kiosks offer over-the-counter service only? The definition describes them as such, but then if there is no variability within the outlet, e.g., all supermarkets are self-service, then it is not clear to me how can you estimate a coefficient for self-service variable for supermarket?
Table 4: To be more specific, please mention in the table title that the results are the marginal effects.
Page 1: “Distance to the outlet from the household location had a negative effect on the use of supermarket with an additional unit increase in distance reducing the probability of supermarket usage by 0.0004 units.” – please specify the units, are they percentage points? The same in the next sentences describing results.
You mentioned “household head income” on the top of the second page, but should it be “household income” only? If it should be “household head income,” it needs to be corrected earlier in the manuscript.
Please use “statistically significant,” not “statistically different.”
“Providing access to credit increased the number of nutrient dense porridge units that could be accessed through vendors by 0.7177.” – please be consistent about how you interpret the coefficients. Previously, they were interpreted using impact on the probability. But here, it is interpreted as impact on the number of nutrient dense porridge units, which is different from the probability. Please correct also in the rest of the manuscript. For example, see another sentence: “The use of self-service strategy increased the units of the nutrient-dense porridge flour that could be purchased through the supermarkets by 1.6317 units and 1.1697 for the vendors.”
“The negative impact that credit access had on the use of supermarkets was expected considering that it is very rare to find supermarkets offering goods directly to consumers on credit.” – I do not think this necessarily explains the negative impact. My understanding is that the interpretation is the opposite, if the supermarkets offer the credit, then the probability of purchase from the supermarket gets smaller.
Please check and correct the numbering of the equations.
Reviewer 4 Report
i really appreciate the work of the authors who revised the manuscript extensively. the current form is very clear to understand as authors removed many ambiguities that were in first draft. Now i have again some suggestions as i gave before.
the map of study area should be inserted in section 2.1 study area, which i was required before.
Now table 1 is corrected and same variables are in both table 1 and table 4.
The columns's title in table 4 are not correctly arranged which made the review difficult. It was difficult to understand which marginal effect belong to which outlet...
There is no need to cite any study in abstract.
